# Clinical Effects of Dietary Supplementation of Lutein with High Bio-Accessibility on Macular Pigment Optical Density and Contrast Sensitivity: A Randomized Double-Blind Placebo-Controlled Parallel-Group Comparison Trial

**DOI:** 10.3390/nu12102966

**Published:** 2020-09-28

**Authors:** Naomichi Machida, Marie Kosehira, Nobuyoshi Kitaichi

**Affiliations:** 1Omnica Co., Ltd. TN Koshikawa Building 5th floor 4--21-7 Itabashi, Itabashiku, Tokyo 173-0004, Japan; 2Graduate School of Pharmaceutical Sciences, Josai University, 1-1 Keyakidai, Sakado-city, Saitama 350-0295, Japan; kosehira@josai.ac.jp; 3Department of Ophthalmology, Health Sciences University of Hokkaido, 1757 Kanazawa, Tobetsu-cho, Ishikari-gun, Hokkaido 061-0293, Japan; nobukita@hoku-iryo-u.ac.jp

**Keywords:** lutein, bioaccessibility, macular pigment, contrast sensitivity, glare sensitivity, age-related macular degeneration

## Abstract

Improvements in macular pigment optical density (MPOD) and contrast sensitivity after administration of 12 mg lutein alone and the timing at which efficacy is observed remain unknown. Therefore, lutein (12 mg), a crystalline formulation, was used in this study, considering its bioaccessibility. This study aimed to determine the effects of lutein administration for 16 weeks on MPOD, contrast sensitivity, and glare sensitivity, and changes in serum lutein levels were determined. The study subjects comprised 59 healthy male and female adults aged 20–69 years. The study diet included a placebo (placebo group) or a diet supplemented with 12 mg of lutein (lutein group). Each study diet was continuously administered for 16 weeks. At weeks 8 and 16, MPOD, contrast sensitivity, glare sensitivity, and serum lutein levels were evaluated. Compared with the placebo group, the lutein group showed significantly improved MPOD, contrast sensitivity, and glare sensitivity at week 16 and significantly increased serum lutein levels at weeks 8 and 16. Continuous administration of lutein for 16 weeks, considering its bioaccessibility, increased MPOD; it made the outlines of visible objects clearer and was effective in inhibiting decreases in visual function caused by glare from light.

## 1. Introduction

Age-related macular degeneration is one of the eye diseases caused by aging; it leads to age-related damage to the macula and central part of the retina as well as symptoms such as darkening, distortion, defects of the center of the visual field, and vision loss. Although this disease is the leading cause of blindness in adults in Europe and the USA, it was considered relatively infrequent in Japan. However, the incidence has significantly increased in recent years because of population aging and Westernization of dietary habits, and it is currently the fourth leading cause of blindness in Japan [1].

High concentrations of xanthophyll carotenoids, such as lutein and zeaxanthin and their metabolites, are present in the retinal macular area [2]. These carotenoids protect the macular fovea from photochemical damage caused by visible light, such as blue light, and reduce the risk of damage [3,4].

However, it has been confirmed that carotenoids in the macula decrease with age, and these carotenoids are found to be decreased in patients with early age-related maculopathy compared with those in healthy elderly individuals [5]. In addition, macular pigment optical density (MPOD) analysis has revealed that MPOD tends to decrease with age [6], indicating that it is important to increase the amount of carotenoids in the macular region to preserve eye health.

Lutein is a plant-derived xanthophyll carotenoid found in green and yellow vegetables as well as yellow flowers, such as marigolds [7]. Xanthophyll carotenoids, such as lutein, cannot be produced in vivo and are supplied through dietary intake. Individuals with high blood lutein levels demonstrate high macular pigment (MP) levels [8]. Low blood levels of high-density lipoprotein (HDL) cholesterol are also associated with low lutein levels in the blood and other tissues [9].

To date, improvements in MPOD and color contrast sensitivity have been reported after administration of 12 mg xanthophylls (10 mg lutein and 2 mg zeaxanthin) for 12 months; however, the time to efficacy onset remains unknown [10]. Administration of 6 or 12 mg lutein alone for 12 weeks increased serum lutein levels in both the groups but improved contrast sensitivity only in the 6-mg lutein dose group; these results were not correlated to lutein intake [11]. In a study in which xanthophyll 22 mg/day was ingested for 12 months, the results of ingestion, MPOD improvement, and C/S improvement were reported to have an accurate association, but efficacy onset time might be suggested in shorter than 12 months because every carotenoid in plasma had reached a plateau at six months [12]. There are also reports that evaluated visual function when zeaxanthin and mesozeaxanthin were ingested together with lutein [13,14,15,16,17,18]. Given the uncertainty regarding the efficacy of lutein, i.e., to evaluate lutein alone without other carotenoids, the present study was conducted to confirm the effects of continuous administration of 12 mg lutein for 8 and 16 weeks on MPOD, contrast sensitivity, and glare sensitivity. The efficacy and time to efficacy onset as well as the correlation between serum lutein and HDL levels were verified.

## 2. Materials and Methods

### 2.1. Study Design and Target Population

A randomized double-blind placebo-controlled parallel-group clinical trial was conducted to assess the effects of lutein administration on visual function. This study was conducted in compliance with the ethical principles of the Helsinki Declaration (adopted in 1964, revised in October 2013) as well as the “Ethical Guidelines for Medical and Health Research Involving Human Subjects (Announcement No. 3 of the Ministry of Education, Culture, Sports, Science and Technology and the Ministry of Health, Labour and Welfare in 2014)” and the “Law Concerning the Protection of Personal Information (May 30, 2003, Law No. 57).” The study protocol was approved by the Institutional Review Board of Nihonbashi Egawa Clinic (13 August 2018). All the subjects were fully informed prior to participating in the study, and written informed consent was obtained. Visual function tests were conducted at the Medicine Evaluation Research Laboratory Center, and safety tests and blood sampling were conducted at the K Medical Office TOC Building Clinic. The study was registered under the University Hospital Medical Information Network (UMIN) Clinical Trials Registry System (UMIN Registration No.: UMIN000033854).

The study subjects were healthy adult males and females aged 20–69 years who were publicly recruited by Huma R & D. The inclusion criteria were individuals who had not undergone myopia correction surgery, who had a corrected visual acuity of 0.7 or better in both the eyes, who could be contacted via a computer or a smartphone, who could discontinue eye drops during the study period, who volunteered to participate in the study, and who provided written consent to participate in the study. In prescreening for the study, visual function tests and a preliminary questionnaire survey were conducted.

The exclusion criteria were as follows:Individuals who were currently receiving some form of drug treatment or outpatient treatment;Individuals who were currently undergoing exercise or diet therapy under the supervision of a physician;Individuals who have a prior history of, are currently experiencing, or present with concomitant hepatic or serious renal, endocrine, cardiovascular, gastrointestinal, pulmonary, hematologic, or metabolic diseases;Individuals who have a prior history of chronic diseases, such as arrhythmia, hepatic disorder, renal disorder, cerebrovascular disorder, rheumatism, diabetes mellitus, dyslipidemia, hypertension, and other chronic diseases or who were being treated for these diseases;Individuals who have a prior history of gastrointestinal surgery (appendicitis allowed);Individuals who have eye diseases other than a refractive error (hyperopia, myopia, and astigmatism);Individuals who have a prior history of drug or food allergies;Pregnant or breastfeeding women and women wishing to become pregnant or breastfeed during the study;Individuals who have received blood transfusions or immunizations within the past 3 months or who plan to receive them during the study period;Individuals who consume health foods, supplements, and pharmaceutical products that may affect visual function on a regular basis;Individuals who have been or are currently being regularly monitored as outpatients at a psychiatric department for psychiatric (e.g., depression) or sleep disorders;Individuals who have a prior history of or are currently being treated for alcohol dependence, drug dependence, or substance abuse;Shift workers such as those who work night shifts;Individuals with extremely irregular lifestyles, such as those with irregular eating and sleeping patterns;Current smokers;Individuals who are currently participating or have participated in other clinical trials within 3 months prior to the date of providing informed consent;Individuals who have difficulties in adhering to entering records for each questionnaire;Other individuals considered inappropriate for inclusion in the trial by the principal investigator.

In a previous report in which lutein and bilberry extracts were administered for 12 weeks and significant improvements in MPOD and contrast sensitivity were observed, the sample size was 20 subjects per group [19]. Considering that the study period in this study was 16 weeks, 60 subjects (30 per group) were selected to prevent the final sample size from being <20.

Randomization was performed by the study food randomization manager who was not involved in the intervention of the trial. The allocation manager assigned subjects to either the placebo or lutein group by stratified randomization according to age, sex, and MPOD. The allocation manager kept the test food assignment table strictly confidential until key opening, and blinding was maintained for all parties except the allocation manager.

### 2.2. Study Diet

The widely used dietary supplement lutein is extracted from marigold (*Tagetes patula* L., *T. erecta* L., or interspecific hybrids) flowers. The dietary lutein formulation used in this study was a marigold pigment, provided by Omnica Co., Ltd., prepared by saponifying lutein esters from T. erecta L. flowers under aqueous phase conditions and mixing lutein-free [(3R, 3’R, 6S)-4,5-didehydro-5,6-dihydro-β, β-carotene-3,3’-diol, CAS: 127-40-2] crystals (HiFil™; Industrial Organica SA, Monterrey, Mexico) at a 76.90% purity by high-performance liquid chromatography (HPLC) with vegetable oils and fats to obtain a 20% lutein-free content. The mean particle size of crystalized lutein was adjusted to 25 μm using a nonporous grindstone-type ultrafine particle grinder. These modifications were used considering the bioaccessibility of lutein.

The two study diets were soft capsules comprising marigold dye preparations (lutein) and soft capsules without lutein (placebo) (Table 1). Titanium dioxide and caramel were added as coatings to the soft capsules, and the lutein and placebo capsules were visually indistinguishable.

### 2.3. Administration Method and Schedule

The administration period in this study was 16 weeks. The subjects were instructed to take two capsules once a day for the study period and to maintain their previous daily life patterns. Tablets were to be taken after breakfast or lunch. Tests were performed at weeks 0, 8, and 16 of administration. On the day of testing, each subject rested without using smartphones or other video display terminal devices for approximately 5–15 min after their arrival, and MPOD and contrast sensitivity measurements were performed.

### 2.4. Test Parameters

Testing was performed at prescreening and at weeks 0 (0 w; on the day of initiation of interventions), 8 (8 w), and 16 (16 w) of administration. Records (status of consumption of the study diet, health status, and adverse events) were filled daily after the intervention, and entries were checked once a week. Safety assessments were performed at weeks 0 and 16 of administration (Table 2).

#### 2.4.1. Serum Lutein Levels

Blood sampling (9 mL) was performed and serum was obtained to measure serum lutein levels. Ethanol (600 μL) was added to 150 μL serum collected in 2-mL centrifuge tubes, and the solution was mixed in a tube mixer and centrifuged at 12,500× *g* for 5 min at 4 °C). Approximately 750 μL supernatant was transferred to another 2-mL centrifuge tube, and 600 μL pure water was added. Subsequently, 600 μL mixture of ethyl acetate/hexane (2:8) was added, mixed in a tube mixer, and centrifuged at 12,500× *g* for 5 min at 4 °C; the upper layer was collected and transferred into a new 2-mL centrifuge tube. This procedure was repeated three times, and approximately 1.8 mL of the organic solvent layer was collected in a centrifuge tube, concentrated, and dried with a nitrogen concentrator that blows nitrogen onto the sample. The precipitates were dissolved in 0.1% (*w*/*v*) dibutylhydroxytoluene and 0.1% (*v*/*v*) triethylamine-containing ethyl acetate:hexane (6:4) solution and centrifuged at 12,500× *g* for 5 min at 4 °C; the supernatant was used as the sample solution. Standard solutions were prepared using a lutein reference standard (ChromaDex, Inc., Los Angeles, CA, USA). To determine lutein levels, 20 μL sample solution and 20 μL standard solution were applied in an HPLC system (Shimadzu Corporation: NexeraX2). The analytical conditions were as follows: GL Science Inertsil SIL-100A column (4.6 mm × 250 mm × 3 μm); column temperature, 30 °C; detection wavelength, 450 nm; mobile phase flow rate, 1.4 mL/min; and ethyl acetate/hexane (60:40) mobile phase.

#### 2.4.2. Macular Pigment Optical Density Test

The MPS II macular pigment screener (Elektron Eye Technology Ltd., Cambridge, UK) was used to measure MPOD. MPS II uses heterochromatic flicker photometry to accurately measure the absorbed amount of blue light in MP. Examinees received the explanation of the test to press the button as soon as a flicker is detected. When the plotted graph follows a downward curve with a clearly defined minimum, MPSII software subsequently calculates the MPOD estimate from the curve and the age-based table. MPOD values are nominal between 0.00 and 1.00. A higher MPOD value means a higher MP level.

#### 2.4.3. Contrast and Glare Sensitivity Tests

Contrast and glare sensitivity measurements were performed using the contrast glare tester CGT-2000 (Takagi Seiko Co., Nagano, Japan). The visual target was a built-in double-ring target. There were six target sizes in a descending order—6.3°, 4.0°, 2.5°, 1.6°, 1.0°, and 0.64°—and the contrast of the target was measured in 14 steps of log contrast sensitivities from 0.190 to 2.149, which were logarithmic values of inverse contrast thresholds [20]. The smaller the log contrast sensitivity, the clearer the difference between light and dark on the target. The subjects pressed a button when the double ring was visible for each viewing target size, and the largest log contrast sensitivity that could be identified was measured.

Measurement conditions for contrast sensitivity included an optical distance of 5 m and background luminance at twilight (10 cd/m^2^), and glare sensitivity was measured from illuminating glare light (100,000 cd/m^2^) around the visual target using the same conditions for measuring contrast sensitivity.

#### 2.4.4. Safety Evaluation

Physical measurements, hematology, blood chemistry, urinalysis, and interviews were performed at weeks 0 and 16 of administration to assess safety.

### 2.5. Statistical Analysis

Summary statistics were calculated for each laboratory value, and statistical analyses were performed using the methods described below. Measurement values are presented as mean and standard deviation.

Unpaired *t*-tests were performed for group comparisons of serum lutein levels, contrast sensitivity, glare sensitivity, and safety assessments. Tests for equal variance were performed using the F-test, and those for unequal variance were performed using Student’s *t*-test for equal variance and Welch’s *t*-test. Group comparisons for MPOD were performed using analyses of covariance with the values at week 0 as a covariate. The level of significance was set at 5% in two-tailed tests. Pearson’s correlation coefficient was used to confirm the correlation between blood HDL and serum lutein levels. We did not consider the issue of test multiplicity, as we referred to the literature that previously evaluated the multiple functionalities of lutein [21]. JMP^®^ 13 (SAS Institute Inc., Cary, NC, USA) was used for statistical analyses.

## 3. Results

### 3.1. Subject Characteristics

The study recruitment period was from 11 August 2018 to 26 August 2018, and the study period was from 12 September 2018 to 11 January 2019. A total of 82 individuals were screened using MPOD and contrast sensitivity measurements, and 62 subjects were eligible.

These 62 eligible subjects comprised 20 men and 42 women. A total of 31 subjects were assigned to each group by the study food randomization manager. Three subjects were excluded from the study because of personal reasons before assessments at week 0, and the remaining 59 subjects completed the study diet consumption, predetermined schedule, and study assessments. The characteristics of the study subjects are presented in Table 3. The flow chart of the subject selection is shown in Figure 1. Among the 59 subjects, 1 was excluded from the analysis for MPOD, contrast sensitivity, and glare sensitivity because of possible effects on test results from the subject having undergone testing at week 16 under conditions that differed from those at the other assessment time points.

### 3.2. Serum Lutein Levels

The lutein group showed significantly higher (*p* < 0.001) serum lutein levels at weeks 8 and 16 than the placebo group (Table 4).

The correlation analysis between serum lutein and blood HDL levels showed that the correlation coefficients at weeks 0 and 16 in the lutein group were 0.566 and 0.586, respectively, indicating a correlation between serum lutein and blood HDL levels (Figure 2).

### 3.3. MPOD

Using the MPOD measured at week 0 as a covariate, significantly higher MPOD values were observed in the lutein group than in the placebo group at week 16 (*p* = 0.040), indicating an improvement. However, no other significant differences were found (Table 4).

### 3.4. Contrast Sensitivity and Glare Sensitivity

At week 16 of administration, the lutein group showed significantly higher contrast sensitivity than the placebo group at target sizes of 6.3° (*p* = 0.0012) and 4.0° (*p* = 0.047). At week 16 of administration, the lutein group showed significantly higher glare sensitivity than the placebo group at the index sizes of 4.0° (*p* = 0.014) and 2.5° (*p* = 0.020). However, no other significant differences were observed (Table 5, Figure 3).

### 3.5. Safety

Physical examination, hematology, blood biochemistry, and urinalysis findings showed that urea nitrogen levels at week 16 were significantly higher in the lutein group (14.86 ± 3.65 mg/dL) than in the placebo group (12.93 ± 3.19 mg/dL) (*p* = 0.035). However, there was only a minor change from the normal ranges, indicating no substantial change from week 0. No significant differences were observed in the other test parameters. Physician interviews and assessments of record entries regarding the onset of adverse events during the study period revealed no adverse events attributable to study diet administration.

## 4. Discussion

Humans can absorb and store various carotenoids that are beneficial to maintain and promote health. Absorbed carotenoids are taken up by chylomicrons in the epithelial cells of the small intestine, released into the lymph, and then transported to the peripheral tissues through the blood to reach the liver. It is thought that these carotenoids are subsequently incorporated into lipoproteins synthesized in the liver and then released back into the blood and transported to various tissues. In plasma, carotenoids are present in each lipoprotein fraction; however, their distribution is not uniform. Nonpolar carotenes and lycopenes are more likely to be distributed in very low-density lipoproteins (VLDLs) and low-density lipoproteins (LDLs), whereas the polar carotenoids—lutein and zeaxanthin—are more likely to be distributed in HDL [22]. Given these properties, high levels of HDL are associated with high blood levels of lutein [23], and the present study also showed a positive correlation between HDL and serum lutein levels.

StARD3, a member of the steroidogenic acute regulatory domain family of proteins, is a xanthophyll-binding protein with high affinity to lutein. It is highly expressed in the retina and the center of macula [24], whereas lutein and zeaxanthin and their metabolites accumulate in the macula from carotenoids transported to the retina. Since MP is a filter of the short-wavelength blue light and is a strong antioxidant [25], lutein accumulation in the macular region has been reported to inhibit age-related macular degeneration [26]. In the present study, serum lutein levels in the lutein group were significantly higher than those in the placebo group at weeks 8 and 16. The significantly improved MPOD at week 16 suggested that lutein administration increased the serum level and subsequent accumulation in the macular area of the retina.

Contrast sensitivity is the measurement of the minimum threshold at which the difference in luminance between an object to be viewed and its background can be identified when this difference is reduced. Therefore, contrast sensitivity indicates the ability to distinguish patterns that have no clear contours and have few differences in shading. Thus, this test is considered extremely useful for assessing the daily visual function that cannot be measured by visual acuity tests [27,28]. In addition, the measurement of contrast sensitivity (glare sensitivity) when glare light is delivered from the peripheral visual field in dim vision allows the assessment of decreases in visual function when viewing glare light, such as that when directly viewing a headlight of an opposing car at night [20]. This study demonstrated significant improvements in contrast and glare sensitivities in the lutein group at week 16. MP filters blue light in front of photoreceptors, thereby attenuating the effects of color aberration and light scattering and enhancing visual function [25]. Since there is controversy over the basis of the contribution of MP, it is necessary to draw a conclusion carefully [29,30]. In the present study, lutein administration increased MP, which alleviated the adverse effects of glare disorders, light scattering, and color aberrations [10,31], thereby improving the sensitivity of discriminating targets with blurred contours and inhibiting the decrease in visual function caused by a glare from light.

In a study by Ma et al. [11], lutein alone was administered at a dose of 6 or 12 mg for 12 weeks, and increased serum lutein levels were observed in both the groups; however, contrast sensitivity only improved in the 6-mg dose group. This may be because of decreases in xanthophyll concentrations in the macular region with age [5], indicating that decreases in contrast sensitivity may be observed in the 22–30-year age group of subjects included in Ma et al.’s study possibly because of decreases in MPOD. Moreover, baseline values were significantly higher in the placebo group than in the 12-mg group. Therefore, effective amounts of lutein to be administered could not be identified in that study. Yao et al. [21] reported that 20 mg lutein administered for 12 months significantly increased MPOD at 6 and 12 months; contrast sensitivity at 3, 6, and 12 months; and glare sensitivity at 12 months compared with the placebo. However, in that study, the respective values increased and improved at week 16. Since the lutein used in this study was a crystalline formulation, considering its bioaccessibility, lutein was possibly absorbed more easily than the placebo, thereby allowing lutein to be distributed to the eye in a short period. These findings indicated that the amounts administered as well as the properties of lutein in the formulation are important factors for its absorption and distribution.

The results of this study suggest that MPOD, contrast sensitivity, and glare sensitivity significantly increase with administration of 12 mg of lutein for 16 weeks, indicating improved visual function between weeks 8 and 16.

## 5. Conclusions

Continuous intake of lutein (12 mg) for 16 weeks as a formulation, considering its bioaccessibility, increased MPOD, improved contrast sensitivity, and prevented decreases in visual function caused by a glare from light.

## Figures and Tables

**Figure 1 nutrients-12-02966-f001:**
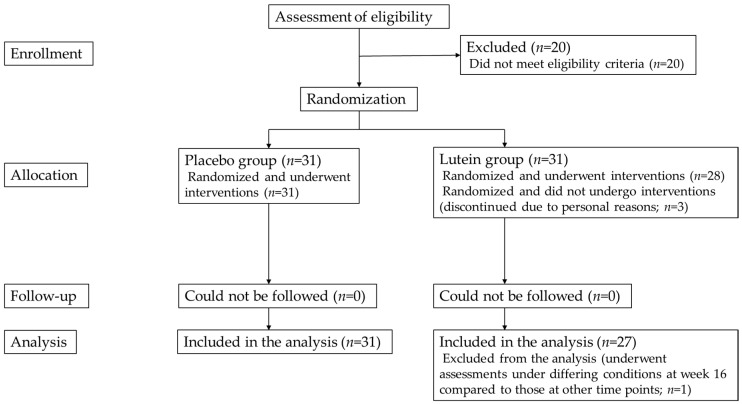
Flowchart of selection of study subjects.

**Figure 2 nutrients-12-02966-f002:**
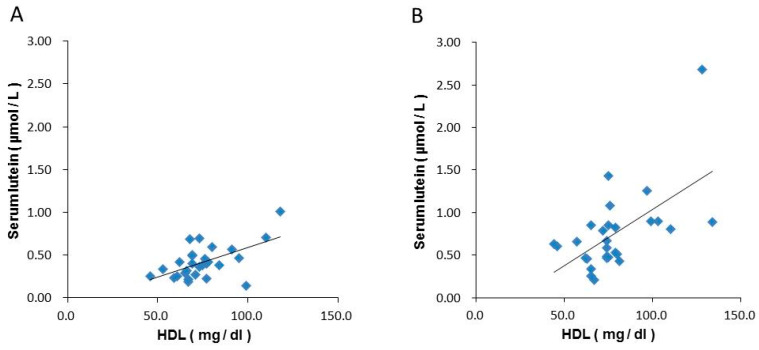
Correlations between blood HDL and serum lutein concentrations in the lutein group. (**A**): 0 w, (**B**): 16 w. HDL; High density lipoprotein cholesterol.

**Figure 3 nutrients-12-02966-f003:**
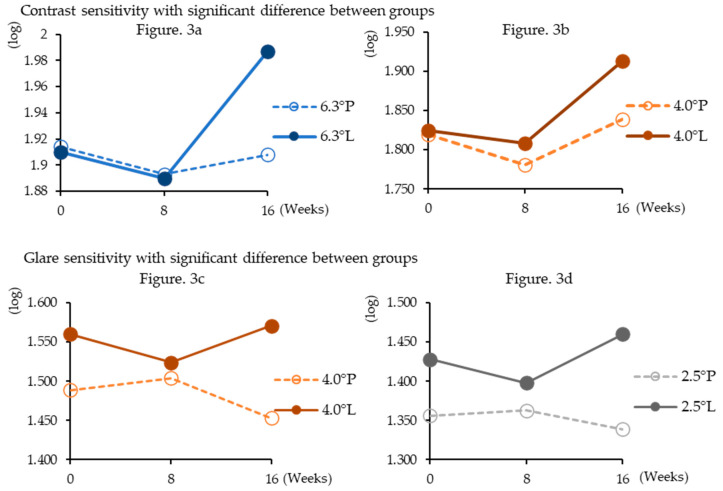
Changes in contrast and glare sensitivities during supplementation. Shows only significantly higher results in the lutein group than in the placebo group. The definition of the legend consists of target sizes and dosing groups (P; placebo group, L; lutein group).

**Table 1 nutrients-12-02966-t001:** Composition of study diet (per capsule).

Components	Placebo Capsule	Lutein Capsule
Marigold dye preparation (lutein-free form)	0 mg	30 mg (6 mg)
Vegetable oil	180 mg	150 mg

**Table 2 nutrients-12-02966-t002:**
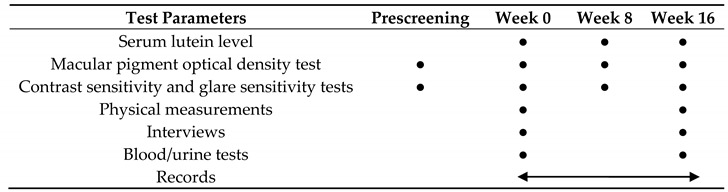
Test parameters and testing schedule.

Closed circles●; Conducted only on that day. Double-headed arrow; Conducted daily during the period.

**Table 3 nutrients-12-02966-t003:** Subject characteristics.

Variables	Placebo Group	Lutein Group
Sex (male/female) (n)	31 (10/21)	28 (9/19)
Age (years)	41.10 ± 12.77	42.61±14.62
Height (cm)	164.26 ± 8.15	162.23±8.35
Body weight (kg)	58.00 ± 10.01	55.10±7.00
Body mass index (kg/m^2^)	21.34 ± 2.05	20.90±1.87

Values are presented as mean ± standard deviation.

**Table 4 nutrients-12-02966-t004:** Changes in serum lutein levels and macular pigment optical density (MPOD) values after supplementation.

Test Parameter	Group	Week 0	Week 8	Week 16
Serum lutein concentration (µmol/L)	Placebo (*n* = 31)	0.381 ± 0.194	0.407 ± 0.187	0.391 ± 0.175
	Lutein (*n* = 28)	0.417 ± 0.193	0.836 ± 0.446 ***	0.751 ± 0.473 ***
MPOD	Placebo (*n* = 62)	0.522 ± 0.145	0.521 ± 0.148	0.523 ± 0.159
	Lutein (*n* = 54)	0.498 ± 0.185	0.521 ± 0.180	0.545 ± 0.173 ^†^

Values are shown as means ± standard deviations. “*n*” indicates the number of subjects for serum lutein levels and number of eyes for MPOD. Statistical significance (unpaired *t*-test): *** *p* < 0.001 vs. placebo group. Statistical significance (ANCOVA): ^†^
*p* < 0.05 vs. placebo group.

**Table 5 nutrients-12-02966-t005:** Changes in contrast and glare sensitivities during supplementation.

	0 w		8 w		16 w	
	Placebo Group (*n* = 62)	Lutein Group (*n* = 54)	Placebo Group (*n* = 62)	Lutein Group (*n* = 54)	Placebo Group (*n* = 62)	Lutein Group (*n* = 54)
Contrast sensitivity						
6.3° (log)	1.914 ± 0.176	1.910 ± 0.161	1.893 ± 0.177	1.890 ± 0.190	1.908 ± 0.172	1.987 ± 0.062 **
4.0° (log)	1.820 ± 0.235	1.825 ± 0.217	1.781 ± 0.212	1.808 ± 0.198	1.839 ± 0.237	1.913 ± 0.155 *
2.5° (log)	1.619 ± 0.308	1.650 ± 0.264	1.596 ± 0.255	1.645 ± 0.268	1.630 ± 0.320	1.671 ± 0.226
1.6° (log)	1.259 ± 0.350	1.299 ± 0.290	1.292 ± 0.327	1.293 ± 0.267	1.291 ± 0.354	1.335 ± 0.267
1.0° (log)	0.985 ± 0.336	1.016 ± 0.287	1.009 ± 0.300	0.986 ± 0.274	0.999 ± 0.368	1.033 ± 0.259
0.64° (log)	0.676 ± 0.330	0.703 ± 0.282	0.702 ± 0.332	0.684 ± 0.327	0.698 ± 0.344	0.703 ± 0.270
Glare Sensitivity						
6.3° (log)	1.521 ± 0.286	1.579 ± 0.202	1.540 ± 0.251	1.523 ± 0.243	1.523 ± 0.292	1.585 ± 0.192
4.0° (log)	1.489 ± 0.288	1.560 ± 0.228	1.504 ± 0.256	1.524 ± 0.247	1.453 ± 0.299	1.571 ± 0.202 *
2.5° (log)	1.356 ± 0.306	1.428 ± 0.208	1.363 ± 0.255	1.398 ± 0.240	1.339 ± 0.322	1.460 ± 0.220 *
1.6° (log)	1.113 ± 0.367	1.177 ± 0.261	1.141 ± 0.328	1.154 ± 0.267	1.091 ± 0.369	1.146 ± 0.246
1.0° (log)	0.808 ± 0.377	0.824 ± 0.278	0.825 ± 0.384	0.805 ± 0.322	0.788 ± 0.419	0.810 ± 0.272
0.64° (log)	0.561 ± 0.336	0.569 ± 0.263	0.563 ± 0.332	0.524 ± 0.303	0.510 ± 0.371	0.536 ± 0.261

Values are presented as means ± standard deviations. “*n*” indicates number of eyes. Statistical significance (unpaired *t*-test); * *p* < 0.05, ** *p* < 0.01 vs. placebo group.

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
