# Peer review of "Clinical Effects of Dietary Supplementation of Lutein with High Bio-Accessibility on Macular Pigment Optical Density and Contrast Sensitivity: A Randomized Double-Blind Placebo-Controlled Parallel-Group Comparison Trial"

_nutrients, 2020, doi:10.3390/nu12102966_

Round 1
Reviewer 1 Report
In this interesting manuscript the authors attempt to show that increased dietary levels of lutein can increase MPOD, contrast sensitivity, and glare sensitivity. There are some problems with this manuscript which need to be corrected before it can be considered.
First, the Methods section lacks some vital details. It specifies that a CGT-2000 contrast glare tester was used but list the log contrast values as ranging from 0.046 to 2.149. These aren't values for this instrument. Are the authors using negative log values? Also, in the statistical analysis section, the authors do not state whether they corrected the false positive rate after having run multiple t-test (e.g. by running a Bonferroni test).
Second, in the Results section, virtually all of the lutein groups had higher contrast and glare sensitivities than the placebo group at time "0w". This higher zero-time baseline could have contributed to the significantly higher results for the lutein group at 16w. The same applies for the glare sensitivity testing and, in addition, the values decreased for almost all placebo groups at 16w, which could have contributed to the significantly higher result in the lutein group. The authors need to address these potential issues for the interpretation of their results. A graph of the results, at least for those groups that showed a statistically significant change, would also be useful.
Author Response
For CGT-2000, logarithmic values were used as described in section "2.4.3. Contrast and glare sensitivity tests". There was an error in the measurement range and measurement stage, so I fixed it. In addition, we have added references.
The literature we referred to in conducting this study did not address the issue of test multiplicity. Therefore, this test does not carry out the test considering the problem of test multiplicity. We have missed the trend of the latest statistical methods. In section “2.5. Statistical analysis”, I added that the references and the problem of test multiplicity were not avoided.
I have created the graph in Table 5 (Figure 3). However, if you graph all the results, it will be huge and difficult to see, so only the results with significant difference between the groups were graphed.
Reviewer 2 Report
Machida et al. report on effects of lutein supplementation on MPOD, contrast sensitivity, and glare disability. The results presented clearly indicate benefit of lutein supplementation to the various parameters tested, and also indicate a strong relationship between serum lutein and serum HDL. Overall, the paper is well written. Below are some points to consider.
- My primary concern with the paper is ignorance of some key previous work in this area. The pioneering work on this topic by Stringham (2003, 2004, 2007, 2008, 2011, 2016, 2017) is conspicuously omitted from the manuscript, and, in the interest of good scholarship, should be examined and considered. The work presented by Machida et al. is primarily a replication / extension of this previous work. Additionally, the work of Nolan and colleagues, most notably their paper on macular carotenoid supplementation in normal subjects (2016: “Enrichment of Macular Pigment Enhances Contrast Sensitivity in Subjects Free of Retinal Disease: Central Retinal Enrichment Supplementation Trials - Report 1.”) is not cited and is directly relevant to the current submission.
- Line 168, the method for measuring MPOD is not adequately described. Were null flicker measures taken in the parafovea as well? Are the values produced by the device true optical density, or simply nominal? Please describe more thoroughly.
- Line 279, “In the present study…” the authors should be careful to state only what they actually found, and not assume effects that were not measured in the study. For example, to say that lutein supplementation and augmented MPOD in the study reduced “light scattering” and “color aberrations” is not valid, because these parameters were not measured. The authors cite previous work on light scattering and color aberrations, but these papers do not address directly the mechanisms for these effects. Moreover, there is controversy in the literature as to effects of wavelength on scatter, and consequently if MPOD can effectively reduce scatter. In fact, Wooten & Geri (1987) showed complete wavelength independence of light scatter in the eye, which renders the contention that MPOD can reduce scatter invalid. Additionally, the presumed effect of reduced chromatic aberration by MPOD on visual resolution was determined to be null by Engles et al. (2007). This is another paper that should be considered by the authors in terms of citation in the review of literature on the topic. My concern here is not with the data presented by Machida et al. – I believe it is valid and interesting – but with stating simply what was found, and maintaining good scholarship (knowledge and appreciation of previous work).
Author Response
1. Thank you for teaching us a pioneering paper. Cited by Stringham on line 55 of section "1. Introduction".
2. Added details of measurement procedure in section "2.4.2. Macular pigment optical density test".
3. Added " Since there is controversy about the basis of the contribution of the MP, it is necessary to draw a conclusion carefully." before the pointed out sentence.
Round 2
Reviewer 1 Report
The manuscript is acceptable in its present form, but a thorough proof-reading is suggested.
Author Response
Thank you for checking.
Corrected the spelling on lines 59-61.
Reviewer 2 Report
Thanks to the authors for making the changes - it improves the manuscript greatly. The only suggestion the I have would be to have someone with a strong background in English carefully inspect the manuscript. For example, in the introduction, within the new edited section, the word "plato" is used instead of "plateau." Also, toward the end of that section, the authors write, "Given the uncertainty regarding the efficacy of lutein...". I believe they mean "lutein alone, without any other carotenoids". Other than those items, I believe the manuscript is ready for publication.
Author Response

(The authors gave the same response as above.)
